# Psychosocial work stressors and mental health in Ph.D. students in Germany— Evidence from two cross-sectional samples

**Meike Heming** [ORCID]*, **Peter Angerer, Mathias Diebig**¤

Institute of Occupational, Social, and Environmental Medicine, Centre for Health and Society, Medical Faculty and University Hospital Düsseldorf, Heinrich-Heine University Düsseldorf, Düsseldorf, Germany

¤ Current address: Department of Work and Organizational Psychology, Faculty I – Psychology, Trier University, Trier, Germany
* Meike.Heming@hhu.de

**Data Availability Statement:** The data that support the findings of this study are available on request from the corresponding author or the Ethics Committee at the Faculty of Medicine of Heinrich

## Abstract

### Background

Ph.D. students have been shown to report a lower mental health status compared to the general population. However, not much is known about the impact of psychosocial work stressors that could contribute to their increased risks of mental health symptoms. This study aims firstly to assess levels of psychosocial stressors, perceived stress, and mental health symptoms in Ph.D. students. Second, it investigates which psychosocial stressors are most strongly associated with mental health symptoms and perceived stress.

### Methods

One self-reported questionnaire was distributed among Ph.D. students at one university in Germany, in summer 2023 (T1) and winter 2023/2024 (T2). Psychosocial stressors were assessed with the short version of the effort-reward imbalance (ERI) questionnaire and with the 16-item DYNAMIK questionnaire. Mental health symptoms were assessed with the 12-item-version of the general health questionnaire (GHQ-12). Perceived stress levels were assessed with the 10-item perceived stress scale (PSS). Within a cross-sectional study design, multiple linear regression analyses were performed in two study samples (n = 267 at T1; n = 244 at T2).

### Results

Ph.D. students reported an imbalance between effort and reward in both study samples (T1: M = 1.34, SD = 0.45; T2: M = 1.27, SD = 0.52). Effort-reward-ratio, boundary permeability, and leader support were associated with mental health symptoms and perceived stress in both study samples, when controlled for age and gender. For example, effort-reward-ratio showed a meaningful impact on mental health symptoms for the study sample at T2 (B = 3.85; p < .05, adj. $R^2$ = 0.288).

Heine University Düsseldorf (ethikkommission@med.uni-duesseldorf.de; subject: 2019-460_5). The data are not publicly available as they contain information that could compromise the privacy of research participants.

**Funding:** The author(s) received no specific funding for this work.

**Competing interests:** The authors have declared that no competing interests exist.

## Discussion

Both study samples showed high prevalence of mental health symptoms and effort-reward imbalance also in comparison to other research findings. An imbalance between effort and reward, boundary permeability and leader support show the most strongly associations with mental health symptoms and perceived stress. Future longitudinal studies could help to support our findings in terms of a causal stressor-strain association. Universities should focus on mental health of Ph.D. students and direct their support towards promoting student-supervisor relationships and clear guidelines for Ph.D. students' working hours.

## Introduction

Common mental disorders are known to be one of the leading causes for sickness absence among employed individuals and almost every third individual in the German population aged 18–79 has a common mental disorder each year [1, 2]. Especially in academia, concerns were raised that employees show a poorer mental health status in comparison to other working populations [3]. For example, teaching staff in universities have shown a similar risk of burnout compared to health care professionals who are known to suffer more from burnout than the average working population [4]. What is neglected in research to this day is the mental health and well-being of Ph.D. students, as one occupational group within academia. Recent research has shown that Ph.D. students report a poorer mental health status (e.g. higher risk of depression or anxiety) and elevated stress levels compared to a general population or to similar aged working professionals [5–8]. Studies among Ph.D. students in Germany and Belgium have shown for example that every third person is at risk of having a depression [9, 10].

Due to the reported elevated levels of stress and poorer mental health status of Ph.D. students, it is important to investigate which factors contribute to this concerning picture. Ph.D. students have to fulfil two different roles: students and researchers. Therefore, they may perceive different magnitudes of challenges and work stressors (i.e. financial aspects, supervisor-student relationship or career opportunities). Ph.D. students have unique and various tasks that are often divided into working on their dissertation project, working on other research projects, teaching and administrative tasks or supervision of students [11]. Thus, investigating the impact of psychosocial stressors on perceived stress and mental health is a crucial first step towards implementation of health-promotion interventions.

In the search for the reasons for this high prevalence of mental health issues of Ph.D. students, it has been suggested that the academic environment might play an important role [12]. A mixed methods systematic review and meta-analysis has shown that isolation and female gender are the strongest risk factors for elevated stress levels in Ph.D. students [7]. In terms of organizational factors, it has been shown that high job demands, low control and conflicts related to work and family are associated with psychological distress in Ph.D. students in Belgium [10]. Job satisfaction/insecurity, life satisfaction and negative institutional support have been associated with depression and perceived stress in Ph.D. students at a university in southwest Germany [9]. In terms of protective factors, it has been shown that a good supervisor-student relationship or an inspirational leadership style of supervisors are important for lower levels of perceived stress [7, 10].

While the above mentioned studies try to identify risk and protective factors for mental health and well-being among Ph.D. students, there are less studies existing that try to assess

these factors within a theoretical framework, in for example work stress models. This could be of help when investigating psychosocial stressors and their association with health outcomes as theoretical models consider key constructs that may directly be associated with health effects [13].

In regard of working conditions for example, some prominent work stress models and standardized questionnaires thereof try to assess stressors in the working environment. The effort-reward imbalance (ERI) model is based on failed reciprocity and assumes that too much effort spent at work, without being sufficiently rewarded for it, can contribute to work stress and poorer health [14]. Assessing the effort-reward (ER-) ratio with its cut-off value of one, can help to distinguish whether individuals are perceiving an imbalance between efforts and rewards [14]. The questionnaires assessing the ERI model (long and short versions) were developed for employed individuals, but today there are modifications existing that try to embed the construct into a university setting for students [15], or into an external setting, where managers can rate their employees efforts and rewards [16].

Using the ERI model in academia to investigate factors which may influence well-being is also suggested by qualitative research [17]. Cross-sectional findings on the ERI model in academia have shown that higher efforts and lower rewards are associated with lower psychological and physical health [18]. Similar findings have been found in a longitudinal design where elevated efforts and a higher over-commitment showed poorer physical and mental health [19].

In Germany, some researchers already investigated the ERI model or combined versions of it among Ph.D. students [20–22]. Recently published findings show that higher effort, lower reward and higher over-commitment are positively associated with perceived stress [20]. A qualitative study explored and supported the use of the ERI model in Ph.D. students as well [21]. A great amount of variance of health satisfaction was explained by rewards [22].

Another more recently developed questionnaire called DYNAMIK, that follows the stressor-strain framework, tries to assess psychosocial stressors in the working context with a focus on digitalization processes that may have changed working environments over the past years [23]. The DYNAMIK questionnaire entails five factors that are recommended to be used in psychosocial risk assessment: workload, boundary permeability, participation, leader support and usability [23]. While some of these components relate to established stress models (such as ERI), the questionnaire is extended by a usability component, as an aspect of a modern work context and by aspects such as working during leisure time or being interrupted at work [23, 24]. These psychosocial stressors were shown to be associated with health outcomes, such as depression, burnout or general health and additional variance to established work stress measures has been shown [23]. However, research findings on the DYNAMIK questionnaire in academia are lacking, so far. This is unfortunate, as the factors of the DYNAMIK questionnaire enable an assessment of Ph.D. students working conditions on a more objective level than the ERI model. This information will help us to find concrete starting points for Ph.D. prevention programs.

In sum, this study aims firstly to assess which psychosocial stressors, levels of perceived stress and mental health problems are present in a sample of Ph.D. students in Germany. Secondly, it is aimed to investigate which psychosocial stressors are associated with perceived stress and mental health symptoms. This study will thereby focus on validated questionnaires that are based on theoretical frameworks assessing psychosocial stressors within the working environment. Assessing psychosocial stressors with the DYNAMIK questionnaire for the first time among Ph.D. students may help to identify psychosocial risks in the workplace of Ph.D. students and thus shed new light on their working conditions and potential starting points for interventions improving mental health and perceived stress for Ph.D. students.

## Methods

### Participants and study design

For this cross-sectional study, Ph.D. students of four faculties at one big university in Germany filled in an online questionnaire. At this university, 30 763 students were enrolled in winter 2023/2024 and 3 562 were enrolled as Ph.D. students. The questionnaire asked about sociodemographic characteristics such as gender, age or progression with the Ph.D. and assessed psychosocial stressors as well as perceived stress and mental health symptoms. Participating faculties were faculty of mathematics and natural sciences, medical faculty, faculty of art and humanities and faculty of business administration and economics. One questionnaire was distributed twice, once during summer 2023 (T1; 23th June to 21th August) and once during winter 2023/2024 (T2; 14th November 2023 to 22th January 2024). The respective faculty distributed the questionnaires via mail towards their enrolled Ph.D. students. Thus, being enrolled as a Ph.D. student at university was an inclusion criteria for the study. There were no other eligibility criteria.

The study was approved by the ethical committee of the medical faculty at the Heinrich-Heine University Düsseldorf (study numbers 2019–460_4 and 2019–460_5). Written informed consent was given by the study participants.

### Measures

The German version of the general health questionnaire (GHQ-12) was used to assess mental health symptoms of the study participants [25]. The GHQ-12 covers questions about mental health symptoms with 12 items and showed good psychometric properties in a general German population when using as a unidimensional measure [26]. Answers are given on a four-point Likert scale from "very rarely" (zero) to "almost always" (three) and participants are asked to refer to their general health status in recent times, in the last two weeks [25]. Different scoring methods are available for the instrument, where often either a bimodal fashion (0-0-1-1) is applied or a sum score range from zero to 36 in a Likert-scale format (0-1-2-3) [26–28]. This study applied the latter scoring method, where higher values represent higher levels of mental health symptoms [27]. To compare our samples in terms of GHQ values with a previous study [10], we additionally coded the items according to the bimodal method (0-0-1-1). With this approach each item is dichotomized reflecting either presence or absence of the respective mental health symptom. We adopted the display of Levecque et al. [10] and show whether at least two (GHQ2+), at least three (GHQ3+) or at least four (GHQ4+) mental health symptoms are present in our study samples. Cronbach's alpha was 0.88 at T1 and 0.88 at T2. Missing values in the GHQ score were replaced by mean value of the score.

The 10-item version of the perceived stress scale (PSS) was used to assess a rather short-term perceived stress level of the past four weeks [29]. The shortened 10-item German version of PSS is a reliable and valid instrument [30]. Items are answered on a five-point Likert scale ranging from "never" (zero) to "very often" (four) [30]. A sum scale was computed, where higher scores reflect higher perceived stress levels ranging from zero to 40. Cronbach's alpha was 0.85 at T1 and 0.85 at T2. Missing values in the PSS score were replaced by mean value of the score.

Psychosocial stressors were assessed with two different instruments. The shortened validated questionnaire version of the effort-reward imbalance (ERI) model was applied within this study [31]. Three items assessed the effort dimension while seven items assessed the reward dimension. Answers were given on a four-point Likert scale from disagreement (one) to agreement (four). Sum scales ranging from one to 12 and from one to 28 are computed for

effort and reward. Higher values represent higher efforts and higher rewards. The ER-ratio was computed according to the formula by Siegrist et al. where effort is divided by reward and multiplied with a correction factor [32]. A value above 1.0 can thereby indicate an imbalance where more efforts are spent than rewards received. Cronbach's alpha was 0.66 for effort and 0.70 for reward at T1. Cronbach's alpha was 0.73 for effort and 0.75 for reward at T2. Missing values in the ER-ratio were replaced by mean value of the ER-ratio.

Psychosocial stressors were also assessed with the validated DYNAMIK questionnaire [23]. The questionnaire assessed 16 items, representing five scales. Four items each assessed work-load and boundary permeability, three items each assessed participation and leader support and usability was assessed with two items. All items are measured on a five-point Likert scale from never/hardly ever or strongly disagree (one) to always or strongly agree (five). All items are recoded in a way that higher values reflect higher levels of the respective stressor. Mean scales ranging from one to five were computed. Cronbach's alpha at T1 for the scales was as followed: 0.71 for workload, 0.73 for boundary permeability, 0.73 for participation, 0.84 for leader support and 0.46 for the two items of the usability scale. Cronbach's alpha at T2 for the scales was as followed: 0.77 for workload, 0.72 for boundary permeability, 0.70 for participation, 0.81 for leader support and 0.62 for the two items for usability.

## Statistical analysis

As only a small number of participants (n = 52) filled in both questionnaires at T1 and T2, the two samples were treated as two cross-sectional samples in analyses. Using two different study samples can help to capture a broader range of experiences and stressors and can thus help to enhance generalizability of findings. Descriptive statistics for the study variables were conducted to observe which psychosocial stressors are present in the study samples at the respective time points and to observe levels of mental health symptoms and perceived stress. Differences on study variables between the two samples at T1 and T2 were investigated by either t-test for independent samples in case of continuous variables or by chi-square test in case of categorical variables. To answer the second research question, multiple linear regression analyses were conducted. Analyses were performed with the mean-imputed analytical samples. Sensitivity analyses were performed for samples with valid data for all study variables (n = 159 at T1 and n = 163 at T2; see supporting information, S1 Table for results on mental health symptoms and S2 Table for results on perceived stress). A stepwise procedure was applied where firstly, age (continuously measured) and gender (male/female) were included into the model as potential confounders. Second, the ER-ratio of the ERI model was included in order to observe and replicate findings of previous studies in academia using the ERI model (e.g. 17, 19, 20). In a third step, the DYNAMIK scales were included to represent psychosocial stressors in modern work environments [23]. The continuously measured outcomes GHQ and PSS were investigated in separate analyses. Presented are unstandardized regression coefficients (B-values) with 95% confidence intervals (CI). In addition, prevalence of mental health symptoms and the ER-ratio are descriptively compared with other published studies. For a statistical power of 0.8, a significance level of 0.05 with a medium effect size and eight predictors, at least 108 participants are required [33]. Analyses were conducted with IBM SPSS Statistics version 28.

## Results

### Study participants

In total, 2.900 Ph.D. students were addressed at T1. Of these, 806 had somehow viewed the questionnaire or started answering, while n = 315 filled in the questionnaire and submitted it

online. At T2, in total 3.070 Ph.D. students were addressed via faculty mails. Of these, n = 740 viewed or started answering the questionnaire, while n = 285 filled in and submitted the questionnaire.

At T1, two cases could not be assigned to a male or female gender and were thus not included in analyses. A closer observation of the missing pattern indicated that few participants had missing values for some of the relevant study questionnaires. Cases where complete questionnaires of interest were missing were excluded. This resulted in n = 305 at T1 and n = 276 at T2. A missing pattern analysis and a conducted Little's Missing Completely at Random (MCAR) Test showed that all of the tested variables were MCAR. Missing values ranged from 0% to 13.3% (i.e. in this case effort variable at T1). Thus, it was decided to perform a mean imputation for three scales in order to keep a sufficient sample size for analysis and to prevent a great amount of information loss. Cases where further missing values were observed for sex, age or other psychosocial stressors were excluded. This resulted in two analytical samples of n = 267 for T1 and n = 244 for T2.

## Descriptive statistics of study samples

The descriptive statistics for the two study samples are shown in Table 1. At both time points, age and gender were similarly distributed among the study samples. About 64% of participants at T1 and 61% at T2 were female. The mean age at both time points was 30 years. Half of the participants were enrolled in a faculty of mathematical and natural sciences. Participants at T1 and T2 showed similar mean values for mental health symptoms and perceived stress levels. Participants showed an imbalance of efforts and reward at both time points, while it was slightly higher for the study sample at T1 (1.34 at T1; 1.27 at T2). Participants at both time points reported elevated levels of psychological stressors such as workload, participation or leader support. The two samples showed significant differences in terms of effort, reward, workload, boundary permeability and leader support with weak effect sizes.

## Associations between psychosocial stressors and mental health symptoms

The results of multiple linear regression analyses estimating associations between psychosocial stressors and mental health symptoms in the two study samples are shown in Table 2. Model 2 shows that an increase in ER-ratio is associated with an increase in reporting increased mental health symptoms (B = 6.13; p < .05 at T2), when adjusted for age and gender. The association between ER-ratio and mental health symptoms is also found in the study sample at T1 with a lower estimated coefficient. With the inclusion of additional psychosocial stressors in Model 3, two more findings are observed in both samples: Increased leader support is associated with less mental health symptoms (B = -1.43; p < .05 at T1) and higher boundary permeability is associated with increased mental health symptoms (B = 2.55; p < .05 at T1). With the inclusion of the additional psychosocial stressors in Model 3, the coefficients for ER-ratio decrease considerably for both study samples. In the study sample at T2, the estimated coefficient for ER-ratio suggests a meaningful impact on mental health status (B = 3.85; p < .05), while the estimated coefficient for ER-ratio at T1 rather suggests a similar trend (B = 1.61; p>.05). Sensitivity analyses for samples with valid data for all study variables showed similar results (see supporting information S1 Table).

## Associations between psychosocial stressors and perceived stress

The results of the multiple linear regression analyses estimating associations between psychosocial stressors and perceived stress in the two study samples are shown in Table 3. Model 2 shows that an increase in ER-ratio is associated with increased levels of perceived stress

**Table 1. Study variables of the two study samples at T1 and T2.**

| | T1 (n = 267) | | | | T2 (n = 244) | | | | p-value[6] | r[7] |
|---|---|---|---|---|---|---|---|---|---|---|
| | n | (%) | Mean | SD[1] | n | (%) | Mean | SD[1] | | |
| **Gender** | | | | | | | | | 0.43 | |
| Female | 171 | 64 | | | 148 | 60.7 | | | | |
| Male | 96 | 36 | | | 96 | 39.3 | | | | |
| **Age** | | | 30.15 | 6.01 | | | 30.35 | 7.0 | 0.73 | |
| (22–87) | | | | | | | | | | |
| **Faculty** | | | | | | | | | 0.07 | |
| (missing at T1 n = 3, at T2 n = 2) | | | | | | | | | | |
| Faculty of Mathematical and Natural Sciences | 136 | 50.9 | | | 124 | 50.8 | | | | |
| Medical Faculty | 41 | 15.4 | | | 49 | 18.4 | | | | |
| Faculty of Arts and Humanities | 73 | 27.3 | | | 49 | 20.1 | | | | |
| Faculty of Business Administration and Economics | 14 | 5.2 | | | 24 | 9.8 | | | | |
| **GHQ[2]** | | | 14.95 | 7.0 | | | 13.78 | 6.81 | 0.06 | |
| Scale range 0–36 | | | | | | | | | | |
| **PSS[3]** | | | 21.19 | 6.18 | | | 20.42 | 6.38 | 0.16 | |
| Scale range 0–40 | | | | | | | | | | |
| **ERI[4] model** | | | | | | | | | | |
| Effort | | | 8.70 | 1.77 | | | 8.19 | 1.97 | 0.00 | 0.13 |
| Scale range 3–12 | | | | | | | | | | |
| Reward | | | 16.50 | 3.39 | | | 17.20 | 3.69 | 0.03 | 0.10 |
| Scale range 7–28 | | | | | | | | | | |
| ER-ratio[5] | | | 1.34 | 0.45 | | | 1.27 | 0.52 | 0.13 | |
| **DYNAMIK** | | | | | | | | | | |
| Scale range 1–5 | | | | | | | | | | |
| Workload | | | 3.47 | 0.66 | | | 3.31 | 0.75 | 0.01 | 0.11 |
| Boundary permeability | | | 3.08 | 0.82 | | | 2.92 | 0.83 | 0.03 | 0.09 |
| Participation | | | 3.73 | 0.75 | | | 3.79 | 0.75 | 0.34 | |
| Leader support | | | 3.50 | 0.94 | | | 3.69 | 0.89 | 0.01 | 0.11 |
| Usability | | | 3.56 | 0.71 | | | 3.52 | 0.80 | 0.56 | |

[1] Standard deviation.

[2] General health questionnaire.

[3] Perceived stress scale.

[4] Effort-reward imbalance.

[5] Effort-reward ratio.

[6] p-value of independent t-test for continuous variables or of chi-square test for categorical variables.

[7] effect size r.

(B = 4.63; p < .05 at T1), when adjusted for age and gender. This association between ER-ratio and perceived stress is also found in the study sample at T2, with a slightly higher estimated coefficient. In Model 3, additional psychosocial stressors show associations with perceived stress. Besides ER-ratio, boundary permeability and leader support are associated with perceived stress in both study samples. In the study sample at T1, high levels of participation and usability are also associated with a decrease in perceived stress levels (B = -1.27; p < .05 for participation). Sensitivity analyses for samples with valid data for all study variables showed similar results (see supporting information S2 Table).

**Table 2. Results from multiple linear regression analyses estimating associations between psychosocial stressors and mental health symptoms at T1 (n = 267) and T2 (n = 244).**

| T1 | Model 1 | | Model 2 | | Model 3 | |
|---|---|---|---|---|---|---|
| | B[1] | 95% CI[2] | B | 95% CI[2] | B[1] | 95% CI[2] |
| Age | -.06 | -.21; 0.08 | -.06 | -.19; .08 | -.03 | -.15; .10 |
| Gender | .90 | -.86; 2.66 | .67 | -1.01; 2.35 | .01 | -1.56; 1.58 |
| ER-ratio[3] | | | **4.68** | 2.91; 6.46 | 1.61 | -.36; 3.57 |
| Workload | | | | | -.52 | -2.07; 1.03 |
| Boundary permeability | | | | | **2.55** | 1.37; 3.72 |
| Participation | | | | | -.71 | -1.87; .46 |
| Leader support | | | | | **-1.43** | -2.45; -.42 |
| Usability | | | | | -.87 | -1.96; .22 |
| | Adj. R² = 0.000 | | Adj. R² = 0.089 | | Adj. R² = 0.223 | |
| **T2** | | | | | | |
| Age | .01 | -.12; .13 | -.01 | -.12; .10 | -.03 | -.13; .08 |
| Gender | 1.76 | .01; 3.51 | 1.15 | -.40; 2.71 | 1.11 | -.38; 2.61 |
| ER-ratio[3] | | | **6.13** | 4.67; 7.58 | **3.85** | 2.12; 5.59 |
| Workload | | | | | .56 | -.73; 1.85 |
| Boundary permeability | | | | | **1.80** | .66; 2.94 |
| Participation | | | | | -.32 | -1.47; .84 |
| Leader support | | | | | -.53 | -1.61; .54 |
| Usability | | | | | -.57 | -1.51; .36 |
| | Adj. R² = 0.008 | | Adj. R² = 0.226 | | Adj. R² = 0.288 | |

In bold p-level < .05.

[1] Unstandardized regression coefficient.

[2] Confidence interval.

[3] Effort-reward ratio.

## Discussion

This cross-sectional study investigated which psychosocial stressors are present in two samples of Ph.D. students in Germany and which of these are associated with mental health symptoms and perceived stress. Both study samples showed a high prevalence of mental health symptoms and an imbalance of efforts spent and rewards received. In multiple linear regression analyses results showed that ER-ratio, boundary permeability and leader support were the most important factors associated with mental health symptoms and perceived stress. A high ER-ratio and high boundary permeability were associated with increased mental health symptoms and perceived stress. Reporting high leader support was associated with decreased mental health symptoms and perceived stress.

### Prevalence of mental health symptoms in comparison to other research findings

When prevalence of single items of the GHQ is compared to findings by Levecque et al., who investigated about 3500 Ph.D. students, considerable differences occur [10]. Our study samples showed a prevalence twice as high for the following six symptoms: could not concentrate, not playing a useful role, could not overcome difficulties, not enjoying day-to-day activities, could not face problems and not feeling happy, all things considered. This results in considerably

**Table 3. Results from multiple linear regression analyses estimating associations between psychosocial stressors and perceived stress at T1 (n = 267) and T2 (n = 244).**

| T1 | Model 1 | | Model 2 | | Model 3 | |
|---|---|---|---|---|---|---|
| | B[1] | 95% CI[2] | B[1] | 95% CI[2] | B[1] | 95% CI[2] |
| Age | -.10 | -.23; .02 | -.10 | -.21; .02 | -.06 | -.17; .04 |
| Gender | 1.33 | -.21; 2.87 | 1.10 | -.35; 2.55 | .41 | -.91; 1.74 |
| ER-ratio[3] | | | **4.63** | 3.09; 6.16 | **1.92** | .26; 3.57 |
| Workload | | | | | -.26 | -1.57; 1.04 |
| Boundary permeability | | | | | **1.64** | .65; 2.63 |
| Participation | | | | | **-1.27** | -2.25; -.29 |
| Leader support | | | | | **-1.49** | -2.34; -.63 |
| Usability | | | | | **-.98** | -1.90; -.07 |
| | Adj. $R^2$ = 0.014 | | Adj. $R^2$ = 0.127 | | Adj. $R^2$ = 0.293 | |
| **T2** | | | | | | |
| Age | .03 | -.09; .14 | .01 | -.09; .11 | -.01 | -.10; .09 |
| Gender | **1.79** | .15; 3.43 | 1.28 | -.21; 2.76 | 1.12 | -.24; 2.49 |
| ER-ratio[3] | | | **5.22** | 3.83; 6.61 | **2.35** | .77; 3.94 |
| Workload | | | | | .65 | -.53; 1.83 |
| Boundary permeability | | | | | **1.64** | .59; 2.68 |
| Participation | | | | | -1.06 | -2.12; .00 |
| Leader support | | | | | **-1.20** | -2.18; -.21 |
| Usability | | | | | -.50 | -1.35; .36 |
| | Adj. $R^2$ = 0.011 | | Adj. $R^2$ = 0.191 | | Adj. $R^2$ = 0.323 | |

In bold p-level < .05.

[1] Unstandardized regression coefficient.

[2] Confidence interval.

[3] Effort-reward ratio.

higher scores compared to findings of Levecque et al. [10]. The authors decided to indicate psychological distress, when at least two out of 12 symptoms were present. It was thus shown, that about 50% fulfilled this criterion while in our case about 70% of participants would fulfill this criterion. On the one hand, mental health symptoms of Ph.D. students could vary in their magnitude among different countries. On the other hand, while Ph.D. students in our study participated in 2023/2024 and thus reported their symptoms ten years later than participants in the study by Levecque et al., this may indicate changes in mental health symptoms over the years [10]. In support of both mentioned explanations, another study among doctoral researchers in UK showed a prevalence of 70.9% for mild to severe depressive symptoms, assessed with the Patient Health Questionnaire (PHQ-9) and a prevalence of 74.2% for anxiety symptoms, assessed with generalized anxiety disorder questionnaire (GAD-7) [8]. However, a recent cross-sectional study among Ph.D. students of one university in the southwest of Germany reported a mean sum score for PHQ-2 that was below the cut-off for major depression, although one third of the participants did reach the cut-off of being at an increased risk for depression [9].

Similar to us, Friedrich et al. also assessed perceived stress among Ph.D. students [9]. They presented a mean sum score of 7.79 (scale range 0–16) [9], while we showed a mean sum score of about 20 for both study samples (scale range from 0–40), which may indicate similar levels of perceived stress. Comparing our mean sum score to a German study validating the German PSS-10 version in a representative population sample (Mean = 12.57 SD = 6.42) shows that

our two samples of Ph.D. students report considerably greater levels of perceived stress [30]. Thus, one could carefully suggest that perceived stress levels of Ph.D. students in Germany are higher compared to a population sample. This is an important but not surprising finding, as research has shown poorer mental health status or elevated stress levels in Ph.D. students and university staff [3, 7, 34].

## Associations between psychosocial stressors, mental health symptoms and perceived stress

Results on prevalence of mental health issues suggest, that it is of importance to take a closer look at the reported psychosocial stressors being associated with mental health symptoms and perceived stress. Ph.D. students may experience different conditions or stressors during their employment resulting in different magnitudes of effects on mental health and perceived stress. Our analyses showed that the strongest psychosocial factors associated with mental health symptoms and perceived stress were receiving support from a leader and boundary permeability as well as an imbalance of effort and reward.

Reporting to receive high leader support was associated with decreased mental health symptoms and perceived stress levels. Similar to this, Levecque et al. showed that the style of leadership (i.e. a lack of inspirational leadership) was associated with mental health symptoms [10]. Negative support was also associated with anxiety disorder and depression in another study [9], and is generally reported as a great challenge during a Ph.D. [35]. Especially in Ph.D. students the supervisor-student role seems to be of high relevance and more important than support from academic peers or relatives for completion of the doctorate [36]. An exploratory study among Ph.D. students in Sweden reported that almost 20% of the study participants thought about changing their supervisor but were afraid that this could go along with adverse consequences for their academic future [37]. In addition, a poor relationship with a superior is not only harmful for Ph.D. students, but was also reported in university staff as a predictor of work stress [38].

Boundary permeability was another strong psychosocial stressor associated with increased mental health symptoms and perceived stress levels in the two study samples at hand. This indicated for example that stressors such as working over hours or having difficulties to combine private and working life with each other are associated with reporting increased mental health symptoms in Ph.D. students. For Ph.D. students it is rather common to work long hours, work during weekends or disregard holidays [39, 40]. Ph.D. students more often work overtime compared to postdocs and to doctorate holders employed in firms [40]. Previous studies reported a mean of 12.1 to 12.6 over hours per week due to the discrepancy between actual working hours and contractual working hours [9, 40]. Working overtime can for example result in conflicts between private and working time [39]. Going in line with our findings, others have also shown that work-family conflict is associated with increased psychological distress [10].

Reporting an imbalance of effort spent and reward received was also associated with increased mental health symptoms and perceived stress levels in the study samples at hand. To our knowledge there are only two other studies available that quantitatively assessed ERI within Ph.D. students [20, 22]. Combining the ERI model with the Stressor-Detachment-Model, it was shown that especially rewards and psychological detachment are important for health satisfaction [22]. It was shown that higher effort, lower reward and higher over-commitment were positively associated with perceived stress, assessed by the shortened four-item PSS version [20]. However, that study reported a lower ER-ratio of 1.01 in Ph.D. students of different universities in Germany compared to our reported ER-ratios. The authors stated that

about 600 of the participants reported an ER-ratio above one and emphasized that associations towards mental health are also shown for lower ER-ratios [20]. A potential explanation for this discrepancy may also lie in the survey time period. First, the timing of survey distribution may be of importance, since teaching responsibilities for example may contribute to different stress perceptions throughout the term. In addition, we can already observe in our two samples that ER-ratios vary slightly for the two time points. As there is probably less interaction with students at T1 due to a summer break and thus less teaching responsibilities this may explain the higher imbalance of effort and reward when Ph.D. students may not receive the same amount of reward for their teaching from their students. In addition to this, it can be observed that leader support is only significantly associated with mental health symptoms at T1, during the summer break, but not at T2. Thus, it can be speculated that leader support may only be relevant when additional support (i.e. from students) is lacking. For example, in a sample of university students it was shown that student support is an important factor which can reduce psychological distress [41]. As components of leader support can be entailed in the reward component of the ERI model, this could also account for the lacking association between ER-ratio and mental health symptoms at T1. Second, the study by Vilser et al. took place during the COVID-19 pandemic from April to June 2022 [20]. Although it is known that due to the COVID-19 pandemic mental health problems such as depression or anxiety have increased in the general population [42, 43], it may be possible that this trend continues further, even after the pandemic. For example, certain dissertation projects may have had to be postponed or changed due to the pandemic, which could increase the time pressure on Ph.D. students, forcing them to put more effort into their work. However, as the study by Vilser et al. included several different universities, it could also indicate that Ph.D. students at our participating university perceive specifically different, i.e. higher imbalance of effort and reward. It could further indicate that our invitation to the study may have appealed to more Ph.D. students that feel particularly stressed. If this would be the case, our results may be overestimated. However, research investigating ERI among academic staff showed for example that higher efforts and lower rewards are associated with poorer physical and mental health [18, 19]. Thus, by showing that the ER-ratio in our study samples was associated with mental health symptoms and perceived stress, we contribute to the already existing findings in academia and replicate the associations among Ph.D. students.

Considering the analyses for perceived stress, two more relevant psychosocial stressors emerged only in the study samples at T1. Increased participation as well as usability were associated with decreased perceived stress levels. The scale for participation included questions about having a say in decisions or having a say in how and which tasks are have to be done. Similarly to our findings, job control was a predictor for decreased psychological distress in Ph.D. students in Belgium [10]. This indicates further that having control at work may actually help Ph.D. students to cope with for example increased work demands or tasks, resulting in perceiving less work stress. However, we could not observe similar findings for the study sample at T2 regarding participation. Explanations on this can only be speculative and need further clarification. For example, it may be possible that for participants during the second survey, having the additional teaching responsibility or having potential upcoming exams or deadlines may overshadow the potential effect of participation on stress perception. Another potential explanation would maybe offer the stage of dissertation project. For example, being at the beginning or end of studies could change how participation would impact stress perception.

Usability was associated with decreased perceived stress levels in the study samples at T1. Two items asked about problems when handling devices and usability of devices or software. This finding indicates, that it may help to provide Ph.D. students with sufficient technical

resources and reliable devices so that they perceive lower stress levels. As to our knowledge, no other work stress models incorporate usability of software or devices in their constructs. This could be a first indication of a somewhat newer work stressor that is relevant in the context of digitalization processes in work environments. Usability of technology, such as technical problems or poor usability, were for example associated as potential stressors in a qualitative study [44]. Due to the COVID-19 pandemic, universities have also had to advance and maintain their digitalization processes (i.e. offering online teaching/exams).

## Practical implications

If longitudinal studies could confirm that leader support is a causal influence on Ph.D. mental well-being, universities should offer courses for both Ph.D. students and for supervisors to promote their relationship. As the results showed that introduction of DYNAMIK items reduced the coefficient of ER-ratio, it needs to be mentioned that this may be due to the contextual overlap of leader support and the reward component of the ERI model: In terms of support, leader can influence for example respect, job security, promotion aspects and possibly salary of their employees. Thus, supervisors should have access to training on how to recognize needs of Ph.D. students and how to support them appropriately [36]. Similar to teaching ratings, one could imagine to introduce ratings of supervisor skills that help to monitor quality of supervision during a Ph.D. [42]. In addition, mentoring contracts that help to formulate expectations should always be documented [42]. Within these contracts we suggest to include regulations or guidelines also related to working hours and the need for availability outside of working hours. These mentoring contracts can help to not only set boundaries but also help to put expectations of both parties in writing. Universities should aim to provide their employees with sufficient and reliable technological resources to prevent increasing stress levels. To sum up, interventions are needed that provide a possibility for Ph.D. students to develop skills themselves (i.e. behavior-based prevention in terms of courses or trainings) but also interventions that change and promote working conditions (i.e. setting-based prevention, guidelines supporting work-life balance or supportive structures improving mental health).

## Limitations

There are limitations that have to be considered when interpreting the results of the study at hand. First, generalization of results on to Ph.D. students in other countries is limited as the online survey was only answered by Ph.D. students at one German university. However, research in various countries has shown that mental health symptoms are not specifically occurring in certain countries [6, 8, 10]. We have also aimed to reach a heterogeneous study sample by providing questionnaires in English and German to also include international Ph.D. students that may have different perceptions of psychosocial stressors. In addition, different faculties were invited to reflect various study fields. However, most of the participants were enrolled in the faculty of mathematics and natural sciences. More in depth comparisons of the faculties could provide useful insights on potential differences in psychosocial stressors, perceived stress or mental health prevalence. In order to reduce the risk of selection bias, we have tried to address all enrolled Ph.D. students of the participating faculties at two time points with several reminder and by using two different study samples. In addition, more female than male participants answered our questionnaire. Reporting a female gender is generally associated with mental health problems in research [6, 9, 45]. However, we have tried to consider this aspect by controlling our analyses for gender. By doing so, we could not see significant gender effects in our study samples, similar to another study among Ph.D. students in Germany [20].

Second, the response rate was rather low, although comparable to other studies in the field [22]. Thus, there is a risk that only Ph.D. students participated in the study that were particularly interested in the study topic. If participants were interested because they felt especially stressed or not satisfied with their Ph.D., this could potentially overestimate our results.

Third, as our study is only based on self-report measures, there may be a chance of common method bias [46]. We have tried to reduce the risk of bias by using standardized and validated questionnaires that have been widely used in research. Due to the cross-sectional study design we also cannot know whether pursuing a Ph.D. entails increased risks for heightened stress levels or mental health symptoms due to poor psychosocial work stressors or whether individuals who do a Ph.D. are more susceptible to have increased risks for mental health symptoms [8]. However, the great amount of research among work stress has shown that particular (combinations of) working conditions can increase the risks for adverse mental health outcomes [47–49].

Fourth, we have observed a not sufficient internal reliability for the usability scale in our two samples. Therefore, findings on usability and its association with perceived stress levels need to be interpreted carefully, and should be replicated in future studies to support whether usability of technical devices is an important psychosocial stressor among Ph.D. students resulting in increased perceived stress levels.

## Conclusions

Ph.D. students showed elevated prevalence of mental health symptoms and perceived stress levels. Boundary permeability, leader support and ER-ratio were the most important psychosocial factors associated with mental health symptoms and perceived stress. Using the ERI model among Ph.D. students is considered helpful to investigate associations with mental health and perceived stress. Future studies could use longitudinal study designs to investigate causality of the associations at hand. Universities should provide Ph.D. students with important resources such as reliable technical devices, but also with guidelines concerning working hours. Interventions and support should also focus on improvement of the student-supervisor relationship.

## Supporting information

**S1 Table. Results from multiple linear regression analyses estimating associations between psychosocial stressors and mental health symptoms at T1 (n = 159) and T2 (n = 163) for only valid values.**
(PDF)

**S2 Table. Results from multiple linear regression analyses estimating associations between psychosocial stressors and perceived stress at T1 (n = 159) and T2 (n = 163) for only valid values.**
(PDF)

## Author Contributions

**Conceptualization:** Meike Heming, Peter Angerer, Mathias Diebig.

**Data curation:** Mathias Diebig.

**Formal analysis:** Meike Heming.

**Investigation:** Meike Heming, Mathias Diebig.

**Methodology:** Meike Heming.

**Project administration:** Meike Heming, Mathias Diebig.

**Writing – original draft:** Meike Heming.

**Writing – review & editing:** Meike Heming, Peter Angerer, Mathias Diebig.

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
