## [Decision Letter · Decision Letter 0]

30 Jul 2024

PONE-D-24-15459Psychosocial work stressors and mental health in Ph.D. students in Germany – Evidence from two cross-sectional samplesPLOS ONE

Dear Dr. Heming,

Thank you for submitting your manuscript to PLOS ONE. After careful consideration, we feel that it has merit but does not fully meet PLOS ONE’s publication criteria as it currently stands. Therefore, we invite you to submit a revised version of the manuscript that addresses the points raised during the review process. The peer-review team provided thoughtful and thorough comments on the manuscript, which you should use to guide your revision effort.  The reviewers requested clarifying details on methods and analysis, with the primary concern of specifying the contribution this study makes to the knowledge base despite the small and limited sample size.

Should you choose to revise and resubmit, please submit your revised manuscript by Sep 13 2024 11:59PM. If you will need more time than this to complete your revisions, please reply to this message or contact the journal office at plosone@plos.org. Please include the following items when submitting your revised manuscript:A rebuttal letter that responds to each point raised by the academic editor and reviewer(s). You should upload this letter as a separate file labeled 'Response to Reviewers'.A marked-up copy of your manuscript that highlights changes made to the original version. You should upload this as a separate file labeled 'Revised Manuscript with Track Changes'.An unmarked version of your revised paper without tracked changes. You should upload this as a separate file labeled 'Manuscript'.

We look forward to receiving your revised manuscript.

Kind regards,

Peter G. Roma, Ph.D.

Academic Editor

PLOS ONE

Journal Requirements:

2. In the online submission form, you indicated that "The datasets generated and analyzed during the study are not publicly available due to data protection regulations, but are available from the corresponding author upon reasonable request."

Reviewers' comments:

Reviewer's Responses to Questions

**Comments to the Author**

1. Is the manuscript technically sound, and do the data support the conclusions?

Reviewer #1: Partly

Reviewer #2: Partly

2. Has the statistical analysis been performed appropriately and rigorously? 

Reviewer #1: No

Reviewer #2: No

3. Have the authors made all data underlying the findings in their manuscript fully available?

Reviewer #1: No

Reviewer #2: No

4. Is the manuscript presented in an intelligible fashion and written in standard English?

Reviewer #1: Yes

Reviewer #2: Yes

5. Review Comments to the Author

Reviewer #1: Dear Authors,

Thank you for the opportunity to read your paper, which is dedicated to the critical subject of mental health and psychosocial work stressors among Ph.D. students at a German university.

The introduction is highly readable and has a good structure and flow. The background for undertaking the issue is clear and logical.

However, some minor and major issues should be addressed.

Minor issues:

1. Please add the percentage of Ph.D. student population in academia in the introduction

2. The diversification of Ph.D. students' faculty is low. Nevertheless, even though the highest rate of participants were students of Mathematical and Natural Sciences, there were also representatives of other faculties.

3. Methods: What does it mean "a big university"? [line: 120] Please add the number of students in general and the number of Ph.D. students.

4. Please add the description of measurement time in view of the academic year and Ph.D. students' responsibilities. The first measurement was conducted during the summer break, while the second between November and January. Yet, you report a higher imbalance during T1. How can you explain it?

5. Please add sample size calculation.

6. Please describe how the common biases were controlled.

7. In the discussion, you only report the main associations, which are repeatable for T1 and T2. Please add to the discussion the differences between prediction models at T1 and T2 for Model 3. For example, leader support was associated with mental health at T1 but not at T2. In model 3 for associations of psychosocial stressors and perceived stress, participation and leader support were not significantly related to perceived stress at T2 (unlike at T1). Please explore it in the discussion.

Major issues:

1. The repeated cross-sectional study design is a positive point of this study. However, the biggest issue is the sample representation. Less than 10% of P.D. students participated in this study, and the sample was from only one university. That undermines the results generalization. When you compare it to the Belgium sample, you compare it to the 15 times larger sample. Thus, it is expected that in a larger sample, the results would be flattened. So, unfortunately, the point of this comparison is missing. However, I agree with the Authors that this topic is vital and should be explored. Thus, I do not reject this paper based on a small sample size.

2. In Table 1 and Table 2, a significance of difference test with effect size should be added.

3. Also, in my opinion, the year of study is missing, and it could be an essential variable. Please add to it to the models, if you have such a possibility.

Reviewer #2: Abstract: This is very good. I would also suggest including the percentage variance explained in the results with the p value. In the discussion section the first sentence needs greater clarity e.g. psychosocial stressors of PhD students and mental health symptoms are higher- but compared to what i.e. another sample, or is it that this particular sample has reported greater scores of mental health symptoms and perceived stress? It would also be helpful to give a specific direction for future research e.g. what types of methods should be used and how?

Introduction

Lines 43 to 49: It's not clear why the authors begin by outlining mental health of academic staff. Academic staff and PhD students have very different roles and the narrative flow could be improved here. It would be better to remove the section on academics and focus on the key population (i.e. PhD students).

Line 52. Should this be, 'every third person'?

It would be helpful to comment on the wide literature, outlining poor student health, and to justify why PhD students might experience greater levels of stress compared to other students (undergraduate and postgraduate). For example, do PhD students have greater competing interests and commitments compared to other types of student? What causes stress in PhD students according to the qualitative evidence?

Line 98. Make sure you writing is explicit i.e. 'the DYNAMIK questionnaire entails five factors...'.

Line 117- This is very clear, but I would also specify what the interventions are aimed at i.e. improving wellbeing/mental health.

Methods:

I would recommend splitting the participants section up from the study design. The participants section should focus on inclusion/exclusion criteria and how/where participants were recruited. You should not report how many participants you recruited here. This should be in the results section. The design section will be very short i.e. a note on the cross-sectional nature of the study, and why you completed two cross-sectional studies, as suggested in the title.

You should include a separate procedure section.

It needs to be clearer throughout the manuscript why two studies were conducted and what is meant by this.

Statistical analysis:

Can you outline what the ER-ratio consists of. How was this captured?

Results:

Lines 219-229: This section should be in the discussion. In the results section you should only include a report of your results. However you can compare your results to other studies in the discussion.

Table 2 is not needed as it was not part of your hypotheses to focus on data from another study? This is not clearly discussed in the methods. The focus should be on your own results and reporting the predictors/association between stressors and mental health.

One of the key limitations of your study is that psychosocial stressors will always predict mental health/distress/stress symptoms, because they are measuring the same underlying variables. I think by 'psychosocial stressors', you really mean 'psychosocial variables' (line 240). This makes more sense considering your variables.

The variance explained by the model should be described in the results. This may be useful to identify what is important to report: https://statistics.laerd.com/spss-tutorials/multiple-regression-using-spss-statistics.php

Discussion:

I think a broader focus on the factors predicting/associated with stress are needed in the discussion. The variables explored are generally discussed well. Some sections could be more concise e.g. practical implications.

This paper is very interesting but there are several areas which need clarification.

6. PLOS authors have the option to publish the peer review history of their article (what does this mean?). If published, this will include your full peer review and any attached files.

Reviewer #1: No

Reviewer #2: No

---

## [Author Response · Author response to Decision Letter 0]

26 Aug 2024

I would like to answer on the data availability statement which was raised in the letter of the revision: The data that support the findings of this study are available on request from the corresponding author. The data are not publicly available as they contain information that could compromise the privacy of research participants. 

All answers to the reviewers are uploaded in the document " response to reviewers".

---

## [Decision Letter · Decision Letter 1]

23 Sep 2024

Psychosocial work stressors and mental health in Ph.D. students in Germany – Evidence from two cross-sectional samples

PONE-D-24-15459R1

Dear Dr. Heming,

We’re pleased to inform you that your manuscript has been judged scientifically suitable for publication and will be formally accepted for publication once it meets all outstanding technical requirements.

Kind regards,

Peter G. Roma, Ph.D.

Academic Editor

PLOS ONE

Additional Editor Comments (optional):

Reviewers' comments:

Reviewer's Responses to Questions

**Comments to the Author**

1. If the authors have adequately addressed your comments raised in a previous round of review and you feel that this manuscript is now acceptable for publication, you may indicate that here to bypass the “Comments to the Author” section, enter your conflict of interest statement in the “Confidential to Editor” section, and submit your "Accept" recommendation.

Reviewer #1: All comments have been addressed

2. Is the manuscript technically sound, and do the data support the conclusions?

Reviewer #1: Yes

3. Has the statistical analysis been performed appropriately and rigorously? 

Reviewer #1: Yes

4. Have the authors made all data underlying the findings in their manuscript fully available?

Reviewer #1: No

5. Is the manuscript presented in an intelligible fashion and written in standard English?

Reviewer #1: Yes

6. Review Comments to the Author

Reviewer #1: Dear Authors,

Thank you for your detailed answers and thorough changes in the manuscript. Regarding the availability of the data, I suggest anonymizing it to comply with the PlosOne journal requirements.

7. PLOS authors have the option to publish the peer review history of their article (what does this mean?). If published, this will include your full peer review and any attached files.

Reviewer #1: **Yes: **Dominika Ochnik

---

## [Editor Report · Acceptance letter]

2 Oct 2024

PONE-D-24-15459R1 

PLOS ONE

Dear Dr. Heming, 

I'm pleased to inform you that your manuscript has been deemed suitable for publication in PLOS ONE. Congratulations! Your manuscript is now being handed over to our production team.

Kind regards, 

on behalf of

Dr. Peter G. Roma 

Academic Editor

PLOS ONE